# Heterogenous Effect of Risk Factors on Breast Cancer across the Breast Density Categories in a Korean Screening Population

**DOI:** 10.3390/cancers12061391

**Published:** 2020-05-28

**Authors:** Boyoung Park, Se-Eun Lim, HyoJin Ahn, Junghyun Yoon, Yun Su Choi

**Affiliations:** 1Department of Medicine, Hanyang University College of Medicine, Seoul 04763, Korea; seeunlim1022@gmail.com (S.-E.L.); deoldsong@gmail.com (H.A.); 2Graduate School of Public Health, Hanyang University, Seoul 04763, Korea; cumyluceat@hanyang.ac.kr; 3Department of Preventive Medicine, Hanyang University College of Medicine, Seoul 04763, Korea; 90dbstn@naver.com

**Keywords:** breast cancer, Breast Imaging-Reporting and Data System, breast density, risk factors, heterogeneous effect

## Abstract

We evaluated the heterogeneity of the effect of known risk factors on breast cancer development based on breast density by using the Breast Imaging-Reporting and Data System (BI-RADS). In total, 4,898,880 women, aged 40–74 years, who participated in the national breast cancer screening program in 2009–2010 were followed up to December 2018. Increased age showed a heterogeneous association with breast cancer (1-year hazard ratio (HR) = 0.92, 1.00 (reference), 1.03, and 1.03 in women with BI-RADS density category 1, 2, 3, and 4, respectively; P-heterogeneity < 0.001). More advanced age at menopause increased breast cancer risk in all BI-RADS categories. This was more prominent in women with BI-RADS density category 1 but less prominent in women in other BI-RADS categories (P-heterogeneity = 0.009). In postmenopausal women, a family history of breast cancer, body mass index ≥ 25 kg/m^2^, and smoking showed a heterogeneous association with breast cancer across all BI-RADS categories. Other risk factors including age at menarche, menopause, hormone replacement therapy after menopause, oral contraceptive use, and alcohol consumption did not show a heterogeneous association with breast cancer across the BI-RADS categories. Several known risk factors of breast cancer had a heterogeneous effect on breast cancer development across breast density categories, especially in postmenopausal women.

## 1. Introduction

Breast cancer is the most common cancer in women worldwide, accounting for 24.4% of all types of cancer; moreover, breast cancer is the leading cause of cancer death in women [1]. In Asian countries, breast cancer incidence is lower; however, it has rapidly increased over the past decades. Changes in demographic factors associated with social and economic development, including lifestyle and reproductive risk factors, increase in breast cancer screening, and awareness in the region, have been attributed to this trend [1,2].

Breast density is a non-modifiable risk factor for breast cancer and can be identified through breast cancer screening using a mammogram. High mammographic density increases the risk of breast cancer; women with a breast tissue density ≥ 75% have a 4–5 times higher risk compared with those with a density < 5% [3,4]. The effect of mammographic density on breast cancer risk has been investigated previously. Studies have suggested that breast density is associated with risk factors of breast cancer such as age, reproductive factors, and body mass index (BMI) [5,6,7]. Other studies investigated whether there is an interaction effect between breast density and the known risk factors on breast cancer. It has been shown that the effect of breast density can be increased or decreased by family history, reproductive factors, behavioral factors, and body mass index (BMI) [8,9,10,11,12,13].

Most previous studies that investigated the interaction effect between breast density and other risk factors on breast cancer development considered other risk factors as effect modifiers and breast density as an independent variable of interest [8,9,10,11,12,13]. Considering a preventive approach in treatment, whether the effects of other breast cancer risk factors differ by breast density [14], needs to be assessed. In addition, the subjects of these studies included only women in the US, Japan, or France. Considering the differences in breast cancer epidemiology between countries, the interaction effect between breast density and other risk factors needs to be assessed in other populations.

In Asian countries, known breast cancer risk factors are less prevalent; however, the prevalence of dense breasts is high [6] and screening mammography programs have only started in some countries [15]. Furthermore, the evaluation of the relationship of mammographic density and breast cancer risk, considering its interaction with other risk factors, has been limited. Therefore, in this study, we comprehensively examined the interaction effect between breast density, which is measured during mammographic screening, and various risk factors on breast cancer development by assessing the heterogeneity of the effect of these risk factors based on breast density in a large East Asian population for individualized risk assessment within the screening setting.

## 2. Results

Table 1 presents the distribution of breast cancer risk factors in participants with or without incident breast cancer according to the Breast Imaging-Reporting and Data System (BI-RADS) density category. Among 5,038,851 subjects, after excluding 141,345 women with missing BI-RADS density or with breast implants, 14.8%, 23.5%, 37.8%, and 23.9% of breast cancer incident cases were classified as BI-RADS density category 1, 2, 3, and 4, respectively. The corresponding values in controls were 27.3%, 27.4%, 30.1% and 15.2%.

The associations between risk factors and the development of breast cancer stratified by breast density, and heterogeneity of the association between BI-RADS categories are presented in Table 2. For participants with BI-RADS density category 1, increased age was associated with a decreased risk of breast cancer (HR = 0.92; 95% CI = 0.90–0.93); no association was observed in participants with BI-RADS category 2 and an increased association was observed in participants with BI-RADS category 3 and 4 (HR = 1.03; 95% CI = 1.01–1.04, HR = 1.03; 95% CI = 1.01–1.06, respectively; P-heterogeneity between the four groups < 0.001). Advanced age at menopause increased breast cancer risk in all BI-RADS categories; this was significantly more prominent in participants with BI-RADS density category 1 (age > 51 years at menopause, HR = 1.46; 95% CI = 1.37–1.56) and less prominent in women with other BI-RAD density categories (HR = 1.28, 95% CI = 1.21–1.35; HR = 1.33, 95% CI = 1.25–1.41; and HR = 1.24, 95% CI = 1.12–1.38 for BI-RADS 2, 3, and 4, respectively; P-heterogeneity = 0.009). Other risk factors did not show a heterogeneous association with breast cancer across BI-RADS categories. Advanced age at menarche, post-menopausal status, and no use of hormone replacement therapy after menopause decreased breast cancer risk; a family history of breast cancer increased the risk irrespective of density category. Menopause at the age of 50–51 years compared with that at ≤ 49 years, use of hormone replacement therapy after menopause for ≥ 1 year compared with that for < 1 year, and higher BMI increased the risk of breast cancer in all density categories.

When stratified by menopausal status, a heterogeneous effect of each risk factor on breast cancer based on the four density categories was not observed in premenopausal participants (Table 3). Despite the lack of heterogeneity, there was no association of BMI with breast cancer in participants with BI-RADS category 1 or 2. However, as BMI increased, the risk of breast cancer was also found to increase; this was more prominent in women with BI-RADS density category 4.

In postmenopausal participants, in addition to age at menopause (> 51 years old, shown in Table 2), a family history of breast cancer, BMI ≥ 25 kg/m^2^, and smoking history showed a heterogeneous association with BI-RADS density categories (Table 4). Those with family history of breast cancer showed an increased risk of breast cancer with a HR of 2.08 (95% CI = 1.75–2.50), 1.72 (95% CI = 1.52–2.00), 1.59 (95% CI = 1.39–1.82), 2.22 (95% CI = 1.82–2.70) in BI-RADS category of 1, 2, 3, and 4, respectively (P-heterogeneity = 0.014). Compared with BMI < 18.5 kg/m^2^, those with BMI ≥ 25 kg/m^2^ had an HR of 2.38 (95% CI = 1.70–3.34), 2.36 (95% CI = 1.80–3.09), 1.64 (95% CI = 1.35–1.98), and 1.79 (95% CI = 1.40–2.29) for those in BI-RADS categories 1, 2, 3, and 4, respectively (P-heterogeneity = 0.008). Smoking increased the risk of breast cancer significantly in participants with BI-RADS density category 3 (HR = 1.20; 95% CI = 1.08–1.34); however, smoking did not show an association in other categories (P-heterogeneity = 0.001).

## 3. Discussion

To the best of our knowledge, this is the one of the largest population-based studies that has investigated the heterogeneity of the effect of risk factors on breast cancer development across BI-RADS density categories. Age and age at menopause (> 51 years vs. ≤ 49 years) showed a heterogeneous association with breast cancer across density categories. Among the participants who underwent menopause, a family history of breast cancer, BMI ≥ 25 kg/m^2^, and smoking showed a heterogeneous effect on breast cancer across the BI-RADS density categories. In premenopausal participants, risk factors did not show a heterogeneous association with breast cancer across the BI-RADS density categories.

In the Korean population, age-specific breast cancer incidence increases up to the age of 45–49 years, and then decreases with further increases in age [16]. In the Korean population, age-specific breast cancer incidence increases up to the age of 45–49 years, and then decreases with further increases in age [16]. A higher incidence rate of breast cancer in women aged < 50 years is a distinct pattern in Asian countries where approximately 50% of breast cancers are diagnosed among women under 50 [2]. Considering that the cumulative proportion of women who experience menopause before 50 and 55 years of age is 46.0% and 90.3% [17], premenopausal breast cancer development is more common than post-menopausal breast cancer. Not only among Korean women but also among other Asian women, breast cancer incidence plateaus or decreases above the age of 50, which is in contrast to the continuous increase in age-specific breast cancer incidence rates with increased age among Western women [18]. In this study, > 80% of the participants with BI-RADS density category 1 were aged ≥ 50 years; 70% of the participants with BI-RADS density category 4 were aged 40–49. Considering the age-related decline in breast density [19], the association of age with breast density may reflect the age-specific breast cancer incidence rate in Korea. The result is in line with a previous study that showed that high breast density was related to the aggressiveness of breast tumors, especially in younger women [20].

A previous study suggested that menopause is more strongly associated with breast density than age [21]. Several studies have investigated the association between mammographic density and breast cancer based on menopausal status and hormone replacement therapy. A study suggested that despite the absence of statistical significance, the strength of the association between density and breast cancer was different, with the strongest association in premenopausal women and postmenopausal women using hormone replacement therapy [8,22,23,24]. However, another study showed the opposite result; the effect of mammographic density was not modified by menopausal status or use of hormone replacement therapy [13]. Reciprocity of the interaction means that if A is an effect modifier of B, then B modifies the effect of A [25]; therefore, when the effect of breast density is significantly different based on other factors, we can expect that the associations of the factor would be different based on the breast density. Kerlikowske et al. showed that women with BI-RADS density category 3 and 4 using hormone replacement therapy had a higher risk of breast cancer than those who did not; this association was not observed in women with BI-RADS density category 1 and 2 [23], similar to our result. However, our results did not show a significant heterogeneity. Despite the lower proportion of women who had ever used hormone replacement therapy than that in a previous study in the United States [21], the heterogenous effects were comparable. More advanced age at menopause was associated with dense breasts [6] and an increased risk of breast cancer [26]; however, the association of age at menopause and breast cancer based on breast density has rarely been studied. In Korean women, the current mean age at menopause is lower than that in other countries such as the United Kingdom, Australia, or Japan but shows an increasing trend [17], which may possibly contribute to the increased breast cancer-related burden. In this study, despite a similar association, advanced age at menopause (>51 years) was associated more with breast cancer in women with low breast density than in women with dense breast tissue.

BMI is a well-known risk factor for the development of breast cancer in postmenopausal women because of the aromatization of androgens into estrogens, but not premenopausal breast cancer where increased BMI may have a protective effect [27]. However, in this study, even premenopausal women with dense breasts (BI-RADS density category 3 and 4) had a slightly increased risk of breast cancer with increased BMI. The combined effect of breast density and BMI on breast cancer has been studied well. The suggested mechanism is that obesity-related insulin deregulation and the adipokine-associated inflammatory response may activate proliferation [28]. In this study, in postmenopausal women with BI-RADS density category 1, BMI had a more prominent effect on breast cancer; this effect decreased as the BI-RADS density category increased. If those with BI-RADS density category 4 had a higher baseline risk of breast cancer, then the added effect of BMI might be lower. Previous studies on the relationship of the effect of breast density and BMI showed inconsistent results; more evident interactions were observed in postmenopausal women [29] or premenopausal women due to a higher risk of estrogen receptor-negative cancer [30]. However, other studies did not find a relationship between breast density and BMI in breast cancer [31], irrespective of menopausal status [10]. In both developed and developing countries, the prevalence of obesity is increasing, especially in younger age groups in developing countries [32]. However, in Korea, prevalence of obesity, defined as BMI ≥ 25 kg/m^2^, has not increased in women. Especially in women aged 20–39 and 40–59 years old, the prevalence of obesity has decreased [33], which is different from the trend in other countries [32]. The unique pattern of changes in BMI in Korea might contribute to the differing results.

Several studies have shown that the association between breast density and the risk of breast cancer is stronger in women with a family history of breast cancer than those without a family history [9,11,34]. The heritability of mammographic density within family members has been shown before [35,36]. A recent study suggested that genetic markers in mammographic density such as percent density and dense area show a shared genetic origin and biological pathways with breast cancer [37]. In this study, we observed the heterogeneous effect of a family history of breast cancer based on BI-RADS density category in postmenopausal women. In women with BI-RADS density category 4, those with a family history of breast cancer showed an increased breast cancer risk; however, in women with BI-RADS density category 3, the increment was less. Considering that the direction and strength of associations were comparable in all BI-RADS density categories, the significant P-heterogeneity may be due to the large sample size of the study subjects.

Despite the well-known carcinogenic effect of smoking, there has been a debate on the association between smoking and breast cancer. However, recent studies demonstrated that smoking increased the risk of breast cancer [38], as well as postmenopausal breast cancer risk [39]. In this study, we did not observe an overall increased risk of smoking in all, premenopausal, or postmenopausal women. It could be attributed to the low prevalence of ever-smoking in Korean women in this study’s subjects, which was similar to that reported in previous studies [40,41]. However, in postmenopausal women, a heterogeneous effect of smoking was observed; smoking significantly increased the risk in women with BI-RADS density category 3 and marginally decreased the risk in women with BI-RADS density category 2. Smoking, due to its antiestrogenic effect on breast tissue, is associated with a decrease in breast density [42,43]. Further investigation is required to understand the effects of smoking on breast density and its association with the risk of breast cancer.

The effect of the decrease in risk with increased age at menarche and a non-significant association of oral contraceptive use irrespective of the breast density category were comparable with findings in previous studies [26,44]. Lower use of oral contraceptives in the Korean population may be the cause of the non-significant association [45]. Light drinking is not associated with most types of cancer and only leads to a mild increases in the risk of breast cancer [46]. The non-significant association of alcohol consumption might be attributed to the low alcohol consumption among Korean women [47]. In Korean women, the prevalence of drinking, defined as ≥ 1 drink of alcohol per month during the last year, was around 40% [41] and the average alcohol consumption was 8 g/day [48].

There are several limitations of this study. First, the study population comprised cancer screening examinees and their baseline characteristics may not be comparable with those of non-examinees. Considering that breast cancer screening participation rate in 2009–2010 was approximately 45% [49] and we included all breast cancer examinees, our study covered almost half of the target population. However, considering that women with lower levels of household income or education participate less in breast cancer screening programs [50], a possible selection bias still remains. Secondly, the BI-RADS density classification provides information to physicians regarding the likelihood of missing a lesion that may be masked by dense tissues; BI-RADS does not quantify breast cancer risk exactly [51]. The BI-RADS density categories were reported by radiologists at many different screening units. However, the inter-observer agreement on BI-RADS categories is substantial [52,53] and a mammographic education program has been conducted to standardize the performance of radiologists in Korea [54]. Furthermore, although the automated breast density and BI-RADS categories showed modest agreement, their association with breast cancer was similar [55], suggesting that the HRs in this study are robust. Thirdly, the information regarding all risk factors except BMI were obtained from self-administered questionnaires; thus, there may be a bias in the responses. However, this would lead to non-differential misclassification and the effects on the results would not be significant [56]. Regarding risk factors, due to a lack of information, some important risk factors for breast cancer, such as parity-related factors, could not be considered. Therefore, the residual confounding effect of unaccounted variables or variables with broad categories may affect the results. In addition, several risk factors, such as BMI, oral contraceptive use, or hormone replacement therapy use, can be changed during one’s lifetime; however, we considered these risk factors at a single time point at their screening during the year of 2009–2010. Fourthly, cancer incidence was identified using the ICD-10 code and catastrophic illness registry in the National Health Insurance Service (NHIS) database, which has the potential for misclassification. However, using the catastrophic illness registry is related to the reimbursement of co-payment, requiring relevant clinical information by the insurance administration. In addition, it covers more than 97% of cancer patients in the Korea Central Cancer Registry. Thus, bias regarding the definition of breast cancer incidence would be minimal. When we separated invasive breast cancer and ductal carcinoma in situ, the results were comparable with the original results, supporting the robustness of the results. The follow-up time of ≥8 years would not be enough to identify all breast cancers. Breast cancer risk assessment estimates mostly 5-year risk but long-term risk assessment for 10 years or more is also needed [57]. Thus, the follow-up period of this study represents an intermediate time point. 

Despite these limitations, our study has a number of strengths including the large sample size and prospective design. The follow-up period was > 8 years and enabled us to study the short- and intermediate-term effects of risk factors and breast density on breast cancer. In addition, the study population covers a large proportion of the female population in Korea and various risk factors interacting with breast density were considered together.

## 4. Materials and Methods

### 4.1. Study Population 

The NHIS in Korea supports biennial health examinations for individuals aged ≥ 40 years. It also supports the screening of stomach, liver, colon, breast, and cervical cancer in the eligible population. Participants of health examinations and cancer screening are asked to complete self-reported questionnaires, which collect information on lifestyle factors, family history of chronic diseases and cancer, and reproductive factors. The information in the questionnaires and results of health examinations and cancer screening are collected through the NHIS screening database. All participants provided consent that allowed the transfer of information to the NHIS database.

For breast cancer, NHIS supports biennial mammographic screening for women aged ≥ 40 years. Approximately 40% of women who are eligible participate in the screening; approximately 3 million mammographic screenings are performed each year. During screening, data on the BI-RADS density assessment and BI-RADS assessment categories were collected. Breast cancer incidence was defined via linking the database with the NHIS medical usage database to obtain the information related to either invasive breast cancer (C50) or ductal carcinoma in situ (D05) and catastrophic illness codes up to December 2018. Upon reviewing the study proposal and request to the National Health Insurance Sharing Service, the NHIS database was made available for research. The study proposal was approved by the Institutional Review Board of the Hanyang University College of Medicine (IRB no. HYI-18-175-1).

We identified 5,317,312 women who participated in health examinations and breast cancer screenings between 2009 and 2010. Among these, women aged ≥ 75 and women who had a medical usage record for any type of cancer with catastrophic illnesses code before the date of breast cancer screening or within 3 months from the date of breast cancer screening (*n* = 278,461) were excluded. The remaining 5,038,851 were included in the analysis as subjects. Women who newly acquired medical usage record for breast cancer (C50 and D05) with a catastrophic illness code (*n* = 55,538) by December 2018 were defined as incident breast cancer cases. In addition, 141,345 women with missing information on breast density or with breast implants were excluded; the remaining 4,897,506 women including 54,164 breast cancer cases were included in the analysis for the analysis stratified by BI-RADS density categories.

### 4.2. Mammographic Density

BI-RADS breast density was classified into four categories: 1, almost entirely fat; 2, scattered fibroglandular densities; 3, heterogeneously dense; and 4, extremely dense.

### 4.3. Risk Factors for Breast Cancer

Information on breast cancer risk factors was obtained from self-reported questionnaires. The questionnaire for health examination included smoking history in one’s lifetime, mean days of drinking per week during the last 1 year, and mean days of physical activity during the last one week. BMI was calculated based on the weight and height measured during the health examination. Reproductive factors, including age at menarche, menopausal status, age at menopause, use of hormone replacement therapy, and oral contraceptive, and family history of cancer in first-degree relatives were obtained from the questionnaire for cancer screening.

### 4.4. Statistical Analysis

The distribution of breast cancer risk factors as a whole and according to the BI-RADS density categories was presented as numbers and percentages. The effect of each risk factor on breast cancer was analyzed using the Cox proportional-hazard regression model adjusted for other risk factors. The results are presented as hazard ratio (HR) and 95% confidence intervals (CIs) for the total number of participants and stratified by BI-RADS density categories. The follow-up was considered from the date of breast cancer screening during the year of 2009–2010 up to December 31, 2018, date of death, or date of any cancer incidence (based on which date came first). Incident breast cancer was considered as an event; death, incidence of other types of cancer, and no incidence of cancer were censored. Additionally, the effect of each risk factor on breast cancer and the stratification by BI-RADS breast density categories were evaluated according to the menopausal status. Heterogeneity of the effect of each risk factor across the BI-RADS density categories was assessed using I^2^ statistics. Analyses were conducted using SAS 9.4 (SAS Institute, Cary, NC, USA) and R software (version 3.5.0) during March–December 2019.

## 5. Conclusions

In summary, some known risk factors of breast cancer showed a heterogeneous effect on breast cancer across breast density categories, especially in postmenopausal women. The association of genetic risk factors or family history, known breast cancer risk factors, especially reproductive factors with breast density is under investigation. Among various hypotheses regarding the mechanism of effect of breast density on breast cancer risk, this research focused on the interaction effect. However, studies have suggested the mediation effect of breast density on the association between known risk factors and breast cancer [58,59]. The role of breast density in breast cancer risk in Asian women, including its mediation effect, needs to be considered in future studies.

## Figures and Tables

**Table 1 cancers-12-01391-t001:** Distribution of risk factors of breast cancer in subjects with or without incident breast cancer by mammographic density.

Risk Factor	Total *				BI-RADS 1				BI-RADS 2				BI-RADS 3				BI-RADS 4			
	Case (*n* = 55,538)		Control (*n* = 4,983,313)		Case (*n* = 8002)		Control (*n* = 1,321,826)		Case (*n* = 12,734)		Control (*n* = 1,328,735)		Case (*n* = 20,478)		Control (*n* = 1,456,419)		Case (*n* = 12,950)		Control (*n* = 736,362)	
**Age, Mean (SD)**	51.42(8.61)		53.99(9.55)		58.18(9.02)		59.16(8.79)		54.69(8.53)		55.37(8.82)		49.73(49.68)		50.05(7.68)		46.68(5.96)		46.93(6.27)	
40–44	14,812	26.7%	1,021,194	20.5%	799	10.0%	80,134	6.1%	1912	15.0%	181,841	13.7%	6061	29.6%	422,209	29.0%	5764	44.5%	320,782	43.6%
45–49	10,816	19.5%	767,644	15.4%	671	8.4%	78,169	5.9%	1655	13.0%	163,811	12.3%	4792	23.4%	316,858	21.8%	3516	27.2%	192,962	26.2%
50–54	12,057	21.7%	1,048,644	21.0%	1322	16.5%	195,289	14.8%	2979	23.4%	312,757	23.5%	4956	24.2%	368,041	25.3%	2481	19.2%	146,180	19.9%
55–59	6570	11.8%	629,573	12.6%	1307	16.3%	193,944	14.7%	2226	17.5%	220,726	16.6%	2212	10.8%	156,762	10.8%	639	4.9%	39,648	5.4%
60–64	6051	10.9%	661,677	13.3%	1741	21.8%	278,030	21.0%	2218	17.4%	221,162	16.6%	1515	7.4%	114,718	7.9%	366	2.8%	23,110	3.1%
65–69	3252	5.9%	460,749	9.2%	1204	15.0%	245,385	18.6%	1124	8.8%	135,147	10.2%	676	3.3%	51,565	3.5%	137	1.1%	9202	1.2%
70–74	1980	3.6%	393,832	7.9%	958	12.0%	250,875	19.0%	620	4.9%	93,291	7.0%	266	1.3%	26,266	1.8%	47	0.4%	4478	8.6%
**Age at Menarche**																				
≤ 15	29,644	53.4%	2,225,369	44.7%	3070	38.4%	405,086	30.6%	5907	46.4%	550,020	41.4%	11,810	57.7%	776,838	53.3%	8256	63.8%	447,264	60.7%
16–17	16,767	30.2%	1,674,284	33.6%	2968	37.1%	512,261	38.8%	4279	33.6%	468,669	35.3%	5855	28.6%	443,922	30.5%	3253	25.1%	200,494	27.2%
> 17	7178	12.9%	909,998	18.3%	1679	21.0%	358,948	27.2%	2080	16.3%	263,207	19.8%	2171	10.6%	187,849	12.9%	1019	7.9%	68,442	9.3%
Missing	1949	3.5%	173,662	3.5%	285	3.6%	45,531	3.4%	468	3.7%	46,839	3.5%	642	3.1%	47,810	3.3%	422	3.3%	20,162	2.7%
**Menopausal Status**																				
Premenopause	31,145	56.1%	2,211,172	44.4%	2214	27.7%	264,153	20.0%	4951	38.9%	467,950	35.2%	13,005	63.5%	871,198	59.8%	10,389	80.2%	565,379	76.8%
Postmenopause	23,406	42.1%	2,687,044	53.9%	5652	70.6%	1,037,607	78.5%	7531	59.1%	835,310	62.9%	7121	34.8%	561,211	38.5%	2358	18.2%	160,705	21.8%
Unknown	987	1.8%	85,097	1.7%	136	1.7%	20,066	1.5%	252	2.0%	25,475	1.9%	352	1.7%	24,010	1.6%	203	1.6%	10,278	1.4%
**Age at Menopause**																				
Premenopause	31,145	56.1%	2,211,172	44.4%	2214	27.7%	264,153	20.0%	4951	38.9%	467,950	35.2%	13,005	63.5%	871,198	59.8%	10,389	80.2%	565,379	76.8%
≤ 49	6240	11.2%	849,192	17.0%	1608	20.1%	359,477	27.2%	2038	16.0%	252,046	19.0%	1765	8.6%	161,408	11.1%	654	5.1%	48,709	6.6%
≤ 51	6219	11.2%	739,689	14.8%	1504	18.8%	289,118	21.9%	1987	15.6%	229,085	17.2%	1906	9.3%	151,566	10.4%	623	4.8%	43,333	5.9%
> 51	9157	16.5%	919,834	18.5%	2256	28.2%	342,243	25.9%	3076	24.2%	303,884	22.9%	2783	13.6%	197,492	13.6%	797	6.2%	49,071	6.7%
Unknown	2777	5.0%	263,426	5.3%	420	5.2%	66,835	5.1%	682	5.4%	75,770	5.7%	1019	5.0%	74,755	5.1%	487	3.8%	29,870	4.1%
**Hormone Replacement Therapy**																				
No Usage	17,409	31.3%	2,142,206	43.0%	4518	56.5%	872,119	66.0%	5677	44.6%	653,482	49.2%	5014	24.5%	421,259	28.9%	1649	12.7%	119,807	16.3%
< 1 Year	3764	6.8%	354,520	7.1%	709	8.9%	103,980	7.9%	1169	9.2%	119,573	9.0%	1326	6.5%	93,767	6.4%	436	3.4%	27,864	3.8%
≥ 1 Year	2089	3.8%	175,189	3.5%	394	4.9%	56,850	4.3%	640	5.0%	57,935	4.4%	744	3.6%	43,006	3.0%	257	2.0%	12,118	1.6%
Premenopause	31,145	56.1%	2,211,172	44.4%	2214	27.7%	264,153	20.0%	4951	38.9%	467,950	35.2%	13,005	63.5%	871,198	59.8%	10,389	80.2%	565,379	76.8%
Unknown	1131	2.0%	100,226	2.0%	167	2.1%	24,724	1.9%	297	2.3%	29,795	2.2%	389	1.9%	27,189	1.9%	219	1.7%	11,194	1.5%
**Oral Contraceptive Use**																				
No Usage	44,041	79.3%	3,950,946	79.3%	6170	77.1%	1,035,264	78.3%	9860	77.4%	1,034,792	77.9%	16,355	79.9%	1,167,219	80.1%	10,582	81.7%	604,281	82.1%
< 6 Months	5231	9.4%	458,377	9.2%	766	9.6%	119,482	9.0%	1246	9.8%	127,147	9.6%	1918	9.4%	135,007	9.3%	1176	9.1%	65,825	8.9%
≥ 6 Months	5200	9.4%	481,640	9.7%	910	11.4%	145,296	11.0%	1370	10.8%	140,045	10.5%	1833	9.0%	128,993	8.9%	972	7.5%	54,928	7.5%
Unknown	1066	1.9%	92,350	1.9%	156	1.9%	21,784	1.6%	258	2.0%	26,751	2.0%	372	1.8%	25,200	1.7%	220	1.7%	11,328	1.5%
**Family History (in First Degree Relative)**																				
No	53,744	96.8%	4,906,815	98.5%	7803	97.5%	1,307,783	98.9%	12383	97.2%	1,308,937	98.5%	19,762	96.5%	1,429,547	98.2%	12,450	96.1%	721,919	98.0%
Yes	1794	3.2%	76,498	1.5%	199	2.5%	14,043	1.1%	351	2.8%	19,798	1.5%	716	3.5%	26,872	1.8%	500	3.9%	14,443	2.0%
**BMI (Kg/m^2^)**																				
< 18.5	1170	2.1%	108,320	2.2%	71	0.9%	17,390	1.3%	130	1.0%	19,134	1.4%	396	1.9%	32,893	2.3%	556	4.3%	36,409	4.9%
< 23	23,032	41.5%	1,978,032	39.7%	2008	25.1%	372,024	28.1%	4107	32.3%	465,450	35.0%	9116	44.5%	673,523	46.2%	7302	56.4%	417,550	56.7%
< 25	13,483	24.3%	1,256,999	25.2%	1913	23.9%	338,400	25.6%	3305	26.0%	357,089	26.9%	5225	25.5%	369,801	25.4%	2713	20.9%	155,051	21.1%
> 25	17,836	32.1%	1,638,953	32.9%	4008	50.1%	593,663	44.9%	5188	40.7%	486,839	36.6%	5738	28.0%	379,964	26.1%	2372	18.3%	127,198	17.3%
Unknown	17	0.0%	1009	0.0%	2	0.0%	349	0.0%	4	0.0%	223	0.0%	3	0.0%	238	0.0%	7	0.1%	154	0.0%
**Smoking in Lifetime**																				
Never Smoked	52,452	94.4%	4,736,003	95.0%	7622	95.3%	1,268,071	95.9%	12,091	95.0%	1,263,989	95.1%	19,253	94.0%	1,377,655	94.6%	12,202	94.2%	695,249	94.4%
Ever Smoked	2795	5.0%	221,184	4.4%	329	4.1%	47,225	3.6%	580	4.6%	58,230	4.4%	1127	5.5%	71,451	4.9%	701	5.4%	38,524	5.2%
Missing	291	0.5%	26,126	0.5%	51	0.6%	6530	0.5%	63	0.5%	6516	0.5%	98	0.5%	7313	0.5%	47	0.4%	2589	0.4%
**Mean** **Days of Drinking per We** **ek during The Last 1 Year**																				
No Drinking	42,705	76.9%	3,954,159	79.3%	6738	84.2%	1,135,424	85.9%	10,213	80.2%	1,073,640	80.8%	15,355	75.0%	1,098,610	75.4%	9373	72.4%	536,517	72.9%
≥ 1 Day	12,328	22.2%	984,611	19.8%	1194	14.9%	174,844	13.2%	2387	18.7%	243,220	18.3%	4981	24.3%	347,920	23.9%	3517	27.2%	196,675	26.7%
Missing	505	0.9%	44,543	0.9%	70	0.9%	11,558	0.9%	134	1.1%	11,875	0.9%	142	0.7%	9889	0.7%	60	0.4%	3170	0.4%
**Mean** **Days of Physical Activity per Week**																				
No Activity	29,643	53.4%	2,782,364	55.8%	4703	58.8%	813,001	61.5%	7000	55.0%	743,828	56.0%	10,564	51.6%	765,617	52.6%	6678	51.6%	383,117	52.0%
≥ 1 Day	25,607	46.1%	2,175,000	43.6%	3256	40.7%	501,918	38.0%	5665	44.5%	578,492	43.5%	9825	48.0%	684,647	47.0%	6230	48.0%	351,149	47.7%
Unknown	288	0.5%	25,949	0.5%	43	0.5%	6907	0.5%	69	0.5%	6415	0.5%	89	0.4%	6155	0.4%	42	0.3%	2096	0.3%

* Women with missing Breast Imaging-Reporting and Data System (BI-RADS) density or with breast implants were included only in total.

**Table 2 cancers-12-01391-t002:** Association between known risk factors of breast cancer and breast cancer development by breast density, in consideration of their interaction on breast cancer risk.

Risk Factor	Total			BI-RADS 1			BI-RADS 2			BI-RADS 3			BI-RADS 4			P-heterogeneity
	HR *	95% CI	P	HR *	95% CI	*p*-value	HR *	95% CI	P	HR *	95% CI	P	HR *	95% CI	P	
**Age**																
1 Year Increment	0.93	(0.92–0.93)	< 0.001	0.92	(0.90–0.93)	< 0.001	1.00	(0.98–1.01)	0.512	1.03	(1.01–1.04)	< 0.001	1.03	(1.01–1.06)	0.001	< 0.001
Age at Menarche																
≤ 15	1			1			1			1			1			
16–17	0.88	(0.87–0.9)	< 0.001	0.88	(0.83–0.92)	< 0.001	0.89	(0.85–0.92)	< 0.001	0.89	(0.86–0.92)	< 0.001	0.90	(0.86–0.94)	< 0.001	0.935
> 17	0.77	(0.75–0.79)	< 0.001	0.76	(0.71–0.80)	< 0.001	0.78	(0.74–0.83)	< 0.001	0.79	(0.75–0.83)	< 0.001	0.84	(0.79–0.90)	< 0.001	0.157
**Menopausal Status**																
Premenopause	1			1			1			1			1			
Postmenopause	0.82	(0.78–0.85)	< 0.001	0.87	(0.78–0.96)	0.004	0.91	(0.84–0.98)	0.011	0.82	(0.76–0.88)	< 0.001	0.76	(0.68–0.85)	< 0.001	0.051
**Age at Menopause Among Menopaused Women**																
≤ 49	1			1			1			1			1			
≤ 51	1.18	(1.14–1.22)	< 0.001	1.17	(1.09–1.25)	< 0.001	1.09	(1.02–1.16)	0.009	1.18	(1.10–1.26)	< 0.001	1.10	(0.98–1.23)	0.102	0.285
> 51	1.42	(1.38–1.47)	< 0.001	1.46	(1.37–1.56)	< 0.001	1.28	(1.21–1.35)	< 0.001	1.33	(1.25–1.41)	< 0.001	1.24	(1.12–1.38)	< 0.001	0.009
**Hormone Replacement Therapy**																
No Usage	0.81	(0.78–0.84)	< 0.001	0.83	(0.77–0.9)	< 0.001	0.9	(0.85–0.96)	0.002	0.84	(0.79–0.89)	< 0.001	0.88	(0.79–0.98)	0.018	0.296
< 1 Year	1			1			1			1			1			
≥ 1 Year	1.18	(1.12–1.24)	< 0.001	1.07	(0.94–1.21)	0.312	1.13	(1.03–1.25)	0.012	1.23	(1.12–1.34)	< 0.001	1.34	(1.15–1.57)	0.001	0.091
**Oral Contraceptive Use**																
No Usage	0.99	(0.97–1.02)	0.617	0.97	(0.89–1.04)	0.347	0.98	(0.93–1.04)	0.582	1.00	(0.96–1.05)	0.897	0.99	(0.93–1.05)	0.786	0.902
< 6 Months	1			1			1			1			1			
≥ 6 Months	0.97	(0.94–1.01)	0.189	0.98	(0.89–1.08)	0.634	0.98	(0.91–1.06)	0.620	0.99	(0.93–1.05)	0.674	0.98	(0.90–1.07)	0.629	0.997
**Family History (in First Degree Relative)**																
No	1			1			1			1			1			
Yes	2.00	(1.89–2.08)	< 0.001	2.13	(1.85–2.44)	< 0.001	1.82	(1.64–2.00)	< 0.001	1.89	(1.75–2.04)	< 0.001	1.96	(1.82–2.17)	< 0.001	0.295
**BMI (Kg/m^2^)**																
< 18.5	1			1			1			1			1			
< 23	1.13	(1.06–1.19)	< 0.001	1.23	(0.97–1.56)	0.088	1.28	(1.08–1.53)	0.005	1.14	(1.03–1.26)	0.013	1.16	(1.07–1.27)	0.001	0.685
< 25	1.14	(1.07–1.21)	< 0.001	1.34	(1.05–1.69)	0.017	1.38	(1.15–1.64)	0.001	1.21	(1.09–1.34)	0.001	1.19	(1.09–1.31)	0.001	0.439
> 25	1.23	(1.16–1.30)	< 0.001	1.62	(1.28–2.05)	< 0.001	1.6	(1.34–1.91)	< 0.001	1.31	(1.18–1.45)	< 0.001	1.28	(1.17–1.41)	< 0.001	0.060
**Smoking in Lifetime**																
Never Smoked	1			1			1			1			1			
Ever Smoked	1.01	(0.98–1.06)	0.471	1.05	(0.94–1.17)	0.422	0.98	(0.9–1.07)	0.622	1.09	(1.02–1.16)	0.009	1.00	(0.92–1.08)	0.937	0.188
**Mean** **Days** **of Drinking per We** **ek** **during the Last 1 Year**																
No Drinking	1			1			1			1			1			
≥ 1 Day	0.99	(0.96–1.01)	0.181	0.95	(0.89–1.02)	0.145	0.97	(0.93–1.02)	0.272	0.99	(0.95–1.02)	0.410	1.00	(0.96–1.04)	0.986	0.583
**Mean** **Days** **of Physical Activity per Week**																
None	0.96	(0.94–0.98)	< 0.001	0.96	(0.90–1.02)	0.171	0.98	(0.94–1.02)	0.357	0.97	(0.94–1.00)	0.087	0.97	(0.93–1.02)	0.204	0.955
1–3 Days/Week	1			1			1			1			1			
≥ 4 Days/Week	1.00	(0.97–1.02)	0.675	1.01	(0.94–1.08)	0.771	0.99	(0.94–1.05)	0.822	1.00	(0.96–1.04)	0.787	0.97	(0.92–1.02)	0.183	0.767

HR: hazard ratio; CI: confidence interval; * Adjusted for listed variables in addition to age.

**Table 3 cancers-12-01391-t003:** Association between known risk factors of breast cancer and breast cancer development by breast density in premenopausal women.

Risk Factor	Total			BI-RADS 1			BI-RADS 2			BI-RADS 3			BI-RADS 4			P-heterogeneity
	HR *	95% CI	P	HR *	95% CI	*p*-value	HR *	95% CI	P	HR *	95% CI	P	HR *	95% CI	P	
**Age at Menarche**																
≤ 15	1			1			1			1			1			
16–17	0.89	(0.87–0.92)	< 0.001	0.93	(0.84–1.02)	0.112	0.89	(0.84–0.95)	0.001	0.88	(0.85–0.92)	< 0.001	0.91	(0.87–0.95)	< 0.001	0.597
> 17	0.78	(0.74–0.81)	< 0.001	0.71	(0.62–0.83)	< 0.001	0.75	(0.68–0.83)	< 0.001	0.82	(0.76–0.87)	< 0.001	0.83	(0.76–0.90)	< 0.001	0.144
**Oral Contraceptive Use**																
No Usage	1.02	(0.98–1.06)	0.381	1.09	(0.94–1.26)	0.261	1.05	(0.96–1.16)	0.302	1.00	(0.95–1.06)	0.926	0.99	(0.92–1.06)	0.716	0.548
< 6 Months	1			1			1			1			1			
≥ 6 Months	1.00	(0.95–1.06)	0.977	0.95	(0.78–1.17)	0.649	1.06	(0.93–1.21)	0.396	1.03	(0.95–1.12)	0.527	0.99	(0.90–1.09)	0.776	0.751
**Family History (in First Degree Relative)**																
No	1			1			1			1			1			
Yes	2.04	(1.92–2.17)	< 0.001	2.13	(1.67–2.70)	< 0.001	1.92	(1.64–2.27)	< 0.001	2.04	(1.85–2.22)	< 0.001	1.92	(1.75–2.13)	< 0.001	0.745
**BMI (Kg/m^2^)**																
< 18.5	1			1			1			1			1			
< 23	1.05	(0.98–1.12)	0.173	0.93	(0.67–1.31)	0.694	0.96	(0.76–1.22)	0.735	1.08	(0.96–1.22)	0.197	1.11	(1.01–1.22)	0.031	0.564
< 25	1.03	(0.96–1.11)	0.351	0.95	(0.67–1.33)	0.757	1.01	(0.80–1.28)	0.936	1.10	(0.98–1.25)	0.114	1.14	(1.03–1.26)	0.011	0.641
> 25	1.02	(0.95–1.09)	0.631	0.86	(0.61–1.21)	0.386	1.08	(0.85–1.37)	0.518	1.15	(1.02–1.30)	0.024	1.19	(1.08–1.32)	0.001	0.323
**Smoking in Lifetime**																
Never Smoked	1			1			1			1			1			
Ever Smoked	1.00	(0.96–1.06)	0.886	0.86	(0.70–1.05)	0.139	1.07	(0.95–1.21)	0.242	1.04	(0.96–1.12)	0.329	1.02	(0.93–1.11)	0.742	0.319
**Mean** **Days** **of Drinking per We** **ek** **during the Last 1 Year**																
No Drinking	1			1			1			1			1			
≥ 1 Day	1.00	(0.97–1.03)	0.945	1.03	(0.93–1.14)	0.540	0.99	(0.93–1.06)	0.887	0.98	(0.94–1.02)	0.456	1.01	(0.96–1.05)	0.752	0.682
**Mean** **Days** **of Physical Activity per Week**																
None	0.97	(0.94–1.00)	0.02	0.97	(0.88–1.08)	0.625	0.99	(0.93–1.07)	0.834	0.98	(0.94–1.02)	0.346	0.96	(0.92–1.01)	0.126	0.880
1–3 Days/Week	1			1			1			1			1			
≥ 4 Days/Week	0.98	(0.95–1.01)	0.131	1.00	(0.88–1.13)	0.967	1.03	(0.95–1.12)	0.484	0.98	(0.93–1.03)	0.348	0.94	(0.89–1.00)	0.044	0.332

HR: hazard ratio; CI: confidence interval; * Adjusted for listed variables in addition to age.

**Table 4 cancers-12-01391-t004:** Association between known risk factors of breast cancer and breast cancer development by breast density in postmenopausal women.

Risk Factors	Total			BI-RADS 1			BI-RADS 2			BI-RADS 3			BI-RADS 4			P-heterogeneity
	HR *	95% CI	P	HR *	95% CI	*p*-value	HR *	95% CI	P	HR *	95% CI	P	HR *	95% CI	P	
**Age at Menarche**																
≤ 15	1			1			1			1			1			
16–17	0.86	(0.83–0.88)	< 0.001	0.86	(0.81–0.91)	< 0.001	0.88	(0.84–0.93)	< 0.001	0.88	(0.84–0.93)	< 0.001	0.86	(0.78–0.94)	0.001	0.909
> 17	0.75	(0.73–0.78)	< 0.001	0.76	(0.71–0.81)	< 0.001	0.79	(0.75–0.84)	< 0.001	0.76	(0.71–0.81)	< 0.001	0.85	(0.75–0.95)	0.006	0.333
**Oral Contraceptive Use**																
No Usage	0.94	(0.90–0.98)	0.003	0.90	(0.83–0.99)	0.024	0.94	(0.87–1.01)	0.081	0.98	(0.90–1.06)	0.579	0.97	(0.85–1.12)	0.722	0.553
< 6 Months	1			1			1			1			1			
≥ 6 Months	0.99	(0.94–1.05)	0.726	1.01	(0.90–1.12)	0.932	0.97	(0.88–1.07)	0.545	0.98	(0.89–1.09)	0.737	1.03	(0.85–1.24)	0.792	0.918
**Family History (in First Degree Relative)**																
No	1			1			1			1			1			
Yes	1.89	(1.75–2.04)	< 0.001	2.08	(1.75–2.50)	< 0.001	1.72	(1.52–2.00)	< 0.001	1.59	(1.39–1.82)	< 0.001	2.22	(1.82–2.70)	< 0.001	0.014
**BMI (Kg/m^2^)**																
< 18.5	1			1			1			1			1			
< 23	1.35	(1.20–1.52)	< 0.001	1.56	(1.11–2.19)	0.011	1.77	(1.35–2.33)	< 0.001	1.26	(1.04–1.52)	0.017	1.48	(1.17–1.88)	0.001	0.225
< 25	1.44	(1.27–1.62)	< 0.001	1.77	(1.26–2.49)	0.001	1.94	(1.48–2.55)	< 0.001	1.45	(1.20–1.76)	0	1.53	(1.19–1.96)	0.001	0.333
> 25	1.67	(1.48–1.88)	< 0.001	2.38	(1.70–3.34)	< 0.001	2.36	(1.80–3.09)	< 0.001	1.64	(1.35–1.98)	< 0.001	1.79	(1.40–2.29)	< 0.001	0.008
**Smoking in Lifetime**																
Never Smoked	1			1			1			1			1			
Ever Smoked	1.04	(0.97–1.11)	0.276	1.14	(0.99–1.31)	0.065	0.89	(0.79–1.01)	0.079	1.20	(1.08–1.34)	0.001	0.91	(0.74–1.11)	0.347	0.001
**Mean** **Days** **of Drinking per We** **ek** **during the Last 1 Year**																
No Drinking	1			1			1			1			1			
≥ 1 Day	1.04	(1.00–1.08)	0.062	1.10	(1.01–1.20)	0.027	1.05	(0.98–1.12)	0.175	1.00	(0.94–1.07)	0.900	1.04	(0.93–1.16)	0.509	0.374
**Mean** **Days** **of Physical Activity per Week**																
None	0.94	(0.91–0.97)	0.001	0.95	(0.88–1.02)	0.169	0.96	(0.90–1.02)	0.156	0.96	(0.90–1.02)	0.136	1.03	(0.93–1.15)	0.534	0.643
1–3 Days/Week	1			1			1			1			1			
≥ 4 Days/Week	1.02	(0.98–1.06)	0.380	1.03	(0.94–1.12)	0.537	0.96	(0.90–1.03)	0.297	1.03	(0.97–1.11)	0.349	1.08	(0.96–1.21)	0.220	0.264

HR: hazard ratio; CI: confidence interval; * Adjusted for listed variables in addition to age, age at menopause, and hormone replacement therapy use.

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
