# Peer review of "Heterogenous Effect of Risk Factors on Breast Cancer across the Breast Density Categories in a Korean Screening Population"

_cancers, 2020, doi:10.3390/cancers12061391_

Round 1

Reviewer 1 Report

General comments:

In the manuscript “Heterogeneous effect of risk factors on breast cancer across the breast density categories”, the authors describe a study of whether the effects of well-established risk factors for development of breast cancer differ by level of breast density (measured via BIRADS) in a screening population of Korean women. They present interesting results that suggest that the effect of age on risk for breast cancer differs by level of breast density, with older age being protective against breast cancer for women with the lowest levels of density. With the large sample size of almost 5 million women, this study represents an important contribution to the literature, and tests of effect modification in a sample this large are well-powered. However, the manuscript could be improved by focusing on the differences in this population compared to what has been found regarding breast cancer risk factors and their interactions in prior studies, and also some additional clarification details.

Major comments:

  • In the introduction, when describing previous interaction studies, it would be helpful to make the population clear, as it would be helpful in interpretation of some of the differences observed in this study.
  • It is critical to clarify how this study is different from prior studies. To me the main contributions seem to be the racial/ethnic population, and also how the interaction is modeled? Did previous studies only look at stratified analyses, and here you will test the interaction more formally? Line 48: the phrase ‘considering the preventative point’ is unclear—do you mean in order to get more precise estimates for risk models? It seems here that many of the risk factors are binary, not quantitative.
  • The association between age and breast cancer seems surprising, and particularly by the density category. I think this warrants more discussion. The age pattern in the Korean population is discussed somewhat, but it would be helpful to comment more on how this differs from other populations. It seems surprising that, overall, the age of the cases is less than the controls, and particularly for BIRADS 1, that younger women are more likely to get cancer. Similarly, it also seems surprising that there are more cases in pre-menopausal women as compared to post-menopausal women, but this is not discussed.
  • It would be helpful if the discussion were framed in context of differences in distribution of risk factors across populations; for example, these seem to be different in Asian women vs. Western populations (e.g. BMI). It may be helpful to discuss these population differences in risk factor distributions also in the introduction.
  • Heterogeneity of family history is only observed in post-menopausal women. Can you explain this result? It seems to me that perhaps the association with family history is not heterogeneous, and you need to further qualify these results?
  • Some tables say “P heterogeneity” and some say “Interaction P”. Were risk factor interaction models also fit, in addition to the heterogeneity tests (I2 statistics)? This is unclear in the methods section.
  • How was BIRADS density treated in the Cox regression model? The tables and results sections suggest that it is a four level categorical variable, but line 232 suggests that it was binary with BIRADS 1-2 and 3-4 grouped? This would change the interpretation of the results.
  • In the statistical analysis, it seems like a single Cox regression model was used that included all of the risk factors? How were the continuous variables (e.g. age) modeled? With linear terms? Is the association with age linear? And were all risk factors related to age included in the model together (e.g. menopause), and were there issues with collinearity? Is it possible that the associations observed with age could be due to residual confounding?
  • A cancer event was defined as both invasive and in situ. Did you consider any sensitivity analyses to separate invasive and DCIS?
  • The paper focuses on moderation effects, but the discussion alludes to possible mediation effects (e.g. smoking). Were mediation analyses considered?

Minor comments:

  • Title: add ‘in a Korean screening population’?
  • Line 37: ‘breast density is non-modifiable’, but then go on to discuss risk factors con modify the effect of density in line 44. I suggest different wording.
  • It would be helpful to show distribution of BIRADS (percentages), by case-control status
  • Do you have any demographic information about the non-screened women? Are these women systematically different?
  • Are there any systematic biases in using ICD codes for the cancer definition?
  • The choice of reference category for some risk factors is odd—e.g. family history

Author Response

Thank you for your valuable comments.

We have revised the paper in accordance with your comments. The revisions are summarized below and highlighted in the manuscript. In addition, we have again opted for proofreading service intensively from an academic editing company. We have attached the respective certificate.

General comments:

In the manuscript “Heterogeneous effect of risk factors on breast cancer across the breast density categories”, the authors describe a study of whether the effects of well-established risk factors for development of breast cancer differ by level of breast density (measured via BIRADS) in a screening population of Korean women. They present interesting results that suggest that the effect of age on risk for breast cancer differs by level of breast density, with older age being protective against breast cancer for women with the lowest levels of density. With the large sample size of almost 5 million women, this study represents an important contribution to the literature, and tests of effect modification in a sample this large are well-powered. However, the manuscript could be improved by focusing on the differences in this population compared to what has been found regarding breast cancer risk factors and their interactions in prior studies, and also some additional clarification details.

Response: Thank you for your comment. We revised the manuscript according to your comments below and give a more detailed explanation regarding the differences in this population compared to what has been found regarding breast cancer risk factors and their interactions in prior studies.

Major comments:

In the introduction, when describing previous interaction studies, it would be helpful to make the population clear, as it would be helpful in interpretation of some of the differences observed in this study.

Response: Thank you for your comment. We discuss the study populations of previous interaction studies as follows: 

“In addition, the subjects of these studies included only women in the US, Japan, or France. Considering the differences in breast cancer epidemiology between countries, the interaction effect between breast density and other risk factors needs to be assessed in other populations.” (line 51-53)

It is critical to clarify how this study is different from prior studies. To me the main contributions seem to be the racial/ethnic population, and also how the interaction is modeled? Did previous studies only look at stratified analyses, and here you will test the interaction more formally? Line 48: the phrase ‘considering the preventative point’ is unclear—do you mean in order to get more precise estimates for risk models? It seems here that many of the risk factors are binary, not quantitative.

Response: In our opinion, this study has two points that distinguish it from previous studies. As you mention, this is the first study regarding the interaction effect between breast density and other risk factors in a Korean population with a large number of subjects. Another is that this is the first study to investigate the effects of known breast cancer risk factors by breast density (a non-modifiable factor). From the preventive standpoint, the variable not amenable to intervention is usually regarded as an effect modifier and exposure that can be eliminated or prevented or novel potential risk factors are regarded as independent variables1. Previous studies have considered other known risk factors as effect modifiers and breast density as an independent variable of interest. In this study, we considered breast density as an effect modifier and other risk factors as independent variables of interest. Regarding modelling for the interaction, we focused on whether the heterogeneity of the effect of known risk factors would differ between density categories as the definition is based on the homogeneity or heterogeneity of effects.

To clarify, we included the following comments:

“Most previous studies that investigated the interaction effect between breast density and other risk factors on breast cancer development considered other risk factors as effect modifiers and breast density as an independent variable of interest2-7. Considering a preventive approach in treatment, whether the effects of other breast cancer risk factors differ by breast density (as an effect modifier)1 needs to be assessed.” (line 47-50)

“Therefore, in this study, we comprehensively examined the interaction effect between breast density, which is measured during mammographic screening, and various risk factors on breast cancer development by assessing the heterogeneity of the effect of these risk factors based on breast density in a large East Asian population for individualized risk assessment within the screening setting.” (line 57-61)

The association between age and breast cancer seems surprising, and particularly by the density category. I think this warrants more discussion. The age pattern in the Korean population is discussed somewhat, but it would be helpful to comment more on how this differs from other populations. It seems surprising that, overall, the age of the cases is less than the controls, and particularly for BIRADS 1, that younger women are more likely to get cancer. Similarly, it also seems surprising that there are more cases in pre-menopausal women as compared to post-menopausal women, but this is not discussed.

Response: As per your comment, we provided a more detailed discussion on age-specific breast cancer incidence rate in Asian women, compared with that in Western women as follows:

“In the Korean population, age-specific breast cancer incidence increases up to the age of 45–49 years, and then decreases with further increases in age8. A higher incidence rate of breast cancer in women aged <50 years is a distinct pattern in Asian countries where approximately 50% of breast cancers are diagnosed among women under 509. Considering that the cumulative proportion of women who experience menopause before 50 and 55 years of age is 46.0% and 90.3%10, premenopausal breast cancer development is more common than post-menopausal breast cancer. Not only among Korean women but also among other Asian women, breast cancer incidence plateaus or decreases above the age of 50, which is in contrast to the continuous increase in age-specific breast cancer incidence rates with increased age among Western women11.” (lines 124-133)

It would be helpful if the discussion were framed in context of differences in distribution of risk factors across populations; for example, these seem to be different in Asian women vs. Western populations (e.g. BMI). It may be helpful to discuss these population differences in risk factor distributions also in the introduction.

Response: As per your comment, we added differences in the distribution of risk factors across populations to the Introduction and Discussion as follows:

“In Asian countries, known breast cancer risk factors are less prevalent; however, the prevalence of dense breasts is high12~” (line 54-55)

“Despite the lower proportion of women who had ever used hormone replacement therapy than that in a previous study in the United States13, the heterogenous effects were comparable.” (line 152-154)

“In Korean women, the current mean age at menopause is lower than that in other countries such as the United Kingdom, Australia, or Japan but shows an increasing trend10, which may possibly contribute to the increased breast cancer-related burden.” (line 157-159)

“In both developed and developing countries, the prevalence of obesity is increasing, especially in younger age groups in developing countries14. However, in Korea, prevalence of obesity, defined as BMI ≥25 kg/m2, has not increased in women. Especially in women aged 20–39 and 40–59 years old, the prevalence of obesity has decreased15, which is different from the trend in other countries14. The unique pattern of changes in BMI in Korea might contribute to the differing results.” (line 176-181)

Heterogeneity of family history is only observed in post-menopausal women. Can you explain this result? It seems to me that perhaps the association with family history is not heterogeneous, and you need to further qualify these results?

Response: Thank you for your valuable comment. The HRs of a family history of breast cancer were 0.48 (95% CI=0.40–0.57), 0.58 (95% CI=0.50–0.66), 0.63 (95% CI=0.55–0.72), and 0.45 (95% CI=0.37–0.55) in BI–RADS density category 1, 2, 3, and 4, respectively. (After changing the category, the corresponding values were 2.08 [95% CI=1.75–2.50], 1.72 [95% CI=1.52–2.00], 1.59 [95% CI=1.39–1.82], 2.22 [95% CI=1.82–2.70]). As you commented, based on the HR, the association of family history might not be heterogeneous between density categories. Rather, the statistical heterogeneity between the four groups (P-heterogeneity=0.014) might be due to the large sample size. We clarified this aspect as follows:

“Considering that the direction and strength of associations were comparable in all BI-RADS density categories, the significant P-heterogeneity may be due to the large sample size of the study subjects.” (line 190-192)

Some tables say “P heterogeneity” and some say “Interaction P”. Were risk factor interaction models also fit, in addition to the heterogeneity tests (I2 statistics)? This is unclear in the methods section.

Response: Interaction-P is a typo. We changed interaction-P to P-heterogeneity in Tables 3 and 4. We are sorry for the confusion.

How was BIRADS density treated in the Cox regression model? The tables and results sections suggest that it is a four level categorical variable, but line 232 suggests that it was binary with BIRADS 1-2 and 3-4 grouped? This would change the interpretation of the results.

Response: We are sorry for the confusion. This is a general classification of dense and fatty breast based on the BI-RADS classification. We have deleted this sentence.

In the statistical analysis, it seems like a single Cox regression model was used that included all of the risk factors? How were the continuous variables (e.g. age) modeled? With linear terms? Is the association with age linear? And were all risk factors related to age included in the model together (e.g. menopause), and were there issues with collinearity? Is it possible that the associations observed with age could be due to residual confounding?

Response: Thank you for your valuable comment. We applied a single Cox regression model including all risk factors and age was treated as continuous variable. In Korea, age-specific breast cancer incidence rate decreases after the age 50 and our study population included women aged ≥40. Thus, in the first analysis, we included age as 5-year categorical groups (40–44, 45–49, 50–54, 55–59, 60–64, 65–69, and 70–74) as in the Table 1. In this analysis, the HRs for each age group was 1.02, 1 (ref), 0.82, 0.74, 0.65, 0.50, and 0.36, respectively. Thus, despite the HR of close to 1 in the 40–44 age group relative to the 45-49, we applied a linear model, considering the general trend between age group and breast cancer risk. In addition, we included possible residual confounding effects as follows in the limitations.

“Therefore, the residual confounding effect of unaccounted variables or variables with broad categories may affect the results.” (line 231-232)

A cancer event was defined as both invasive and in situ. Did you consider any sensitivity analyses to separate invasive and DCIS?

Response: We performed a sensitivity analysis, separating invasive breast cancer and DCIS. The results were comparable with the original results. We added the following comment:

“When we separated invasive breast cancer and ductal carcinoma in situ, the results were comparable with the original results, supporting the robustness of the results.” (line 239-241)

The paper focuses on moderation effects, but the discussion alludes to possible mediation effects (e.g. smoking). Were mediation analyses considered?

Response: We focused on the moderation effect of breast density on known risk factors and did not consider any mediation effect. We added this as a future research direction to the Conclusion as follows:

“Among various hypotheses regarding the mechanism of effect of breast density on breast cancer risk, this research focused on the interaction effect. However, studies have suggested the mediation effect of breast density on the association between known risk factors and breast cancer16, 17. The role of breast density in breast cancer risk in Asian women, including its mediation effect, needs to be considered in future studies.” (line 308-312)

Minor comments:

Title: add ‘in a Korean screening population’?

Response: We added “in a Korean screening population” to the title. (line 3-4)

Line 37: ‘breast density is non-modifiable’, but then go on to discuss risk factors con modify the effect of density in line 44. I suggest different wording.

Response: We changed “modified” to “increased or decreased” in line 45.

It would be helpful to show distribution of BIRADS (percentages), by case-control status

Response: We presented the distribution of BI-RADS density categories in the Results as follows:

“Among 5,038,851 subjects, after excluding 141,345 women with missing BI-RADS density or with breast implants, 14.8%, 23.5%, 37.8%, and 23.9% of breast cancer incident cases were classified as BI-RADS density category 1, 2, 3, and 4, respectively. The corresponding values in controls were 27.3%, 27.4%, 30.1%, and 15.2%.” (line 66-69)

Do you have any demographic information about the non-screened women? Are these women systematically different?

Response: Because only information from screened women is included in the NHIS database, it would be impossible to compare characteristics of non-screened women with screened women directly. However, a nationwide survey in Korea showed that women with lower levels of household income and education participated less in cervical or breast cancer screening programs. We mention this in the Discussion as follows:

“However, considering that women with lower levels of household income or education participate less breast cancer screening programs18, a possible selection bias still remains.” (line 217-219)

Are there any systematic biases in using ICD codes for the cancer definition?

Response: In Korea, more than 97% of incident cancer patients are registered to the catastrophic illness registry using the ICD-10 code for cancer. Thus, we considered the bias would be minimal. This point is presented as follows in the Discussion:

“Fourth, cancer incidence was identified using the ICD-10 code and catastrophic illness registry in the NHIS database, which has the potential for misclassification. However, using the catastrophic illness registry is related to the reimbursement of co-payment, requiring relevant clinical information by the insurance administration. In addition, it covers more than 97% of cancer patients in the Korea Central Cancer Registry. Thus, bias regarding the definition of breast cancer incidence would be minimal.” (line 234-239)

The choice of reference category for some risk factors is odd—e.g. family history

Response: We changed reference categories of family history and drinking and re-calculated the hazard ratios. Please refer to revised Table 2-4. The related comments in the manuscript were also changed.

References

  1. Szklo, M.; Nieto, F. J., Epidemiology: beyond the basics. Jones & Bartlett Publishers: 2014.
  2. Yaghjyan, L.; Colditz, G. A.;  Rosner, B.; Tamimi, R. M., Mammographic breast density and breast cancer risk: interactions of percent density, absolute dense, and non-dense areas with breast cancer risk factors. Breast cancer research and treatment 2015, 150 (1), 181-9.
  3. Maskarinec, G.; Nakamura, K. L.;  Woolcott, C. G.;  Conroy, S. M.;  Byrne, C.;  Nagata, C.;  Ursin, G.; Vachon, C. M., Mammographic density and breast cancer risk by family history in women of white and Asian ancestry. Cancer causes & control : CCC 2015, 26 (4), 621-6.
  4. Conroy, S. M.; Woolcott, C. G.;  Koga, K. R.;  Byrne, C.;  Nagata, C.;  Ursin, G.;  Vachon, C. M.;  Yaffe, M. J.;  Pagano, I.; Maskarinec, G., Mammographic density and risk of breast cancer by adiposity: an analysis of four case-control studies. Int J Cancer 2012, 130 (8), 1915-24.
  5. Maskarinec, G.; Dartois, L.;  Delaloge, S.;  Hopper, J.;  Clavel-Chapelon, F.; Baglietto, L., Tumor characteristics and family history in relation to mammographic density and breast cancer: The French E3N cohort. Cancer Epidemiol 2017, 49, 156-160.
  6. Woolcott, C. G.; Koga, K.;  Conroy, S. M.;  Byrne, C.;  Nagata, C.;  Ursin, G.;  Vachon, C. M.;  Yaffe, M. J.;  Pagano, I.; Maskarinec, G., Mammographic density, parity and age at first birth, and risk of breast cancer: an analysis of four case-control studies. Breast cancer research and treatment 2012, 132 (3), 1163-71.
  7. Yaghjyan, L.; Tamimi, R. M.;  Bertrand, K. A.;  Scott, C. G.;  Jensen, M. R.;  Pankratz, V. S.;  Brandt, K.;  Visscher, D.;  Norman, A.;  Couch, F.;  Shepherd, J.;  Fan, B.;  Chen, Y. Y.;  Ma, L.;  Beck, A. H.;  Cummings, S. R.;  Kerlikowske, K.; Vachon, C. M., Interaction of mammographic breast density with menopausal status and postmenopausal hormone use in relation to the risk of aggressive breast cancer subtypes. Breast cancer research and treatment 2017, 165 (2), 421-431.
  8. Jung, K.-W.; Won, Y.-J.;  Kong, H.-J.; Lee, E. S., Cancer Statistics in Korea: Incidence, Mortality, Survival, and Prevalence in 2016. Cancer Res Treat 2019, 51 (2), 417-430.
  9. Youlden, D. R.; Cramb, S. M.;  Yip, C. H.; Baade, P. D., Incidence and mortality of female breast cancer in the Asia-Pacific region. Cancer Biol Med 2014, 11 (2), 101-115.
  10. Park, C. Y.; Lim, J. Y.; Park, H. Y., Age at natural menopause in Koreans: secular trends and influences thereon. Menopause (New York, N.Y.) 2018, 25 (4), 423-429.
  11. Sung, H.; Rosenberg, P. S.;  Chen, W. Q.;  Hartman, M.;  Lim, W. Y.;  Chia, K. S.;  Wai-Kong Mang, O.;  Chiang, C. J.;  Kang, D.;  Ngan, R. K.;  Tse, L. A.;  Anderson, W. F.; Yang, X. R., Female breast cancer incidence among Asian and Western populations: more similar than expected. Journal of the National Cancer Institute 2015, 107 (7).
  12. Sung, H.; Ren, J.;  Li, J.;  Pfeiffer, R. M.;  Wang, Y.;  Guida, J. L.;  Fang, Y.;  Shi, J.;  Zhang, K.;  Li, N.;  Wang, S.;  Wei, L.;  Hu, N.;  Gierach, G. L.;  Dai, M.;  Yang, X. R.; He, J., Breast cancer risk factors and mammographic density among high-risk women in urban China. NPJ breast cancer 2018, 4, 3.
  13. Boyd, N.; Martin, L.;  Stone, J.;  Little, L.;  Minkin, S.; Yaffe, M., A longitudinal study of the effects of menopause on mammographic features. Cancer epidemiology, biomarkers & prevention : a publication of the American Association for Cancer Research, cosponsored by the American Society of Preventive Oncology 2002, 11 (10 Pt 1), 1048-53.
  14. Ng, M.; Fleming, T.;  Robinson, M.;  Thomson, B.;  Graetz, N.;  Margono, C.;  Mullany, E. C.;  Biryukov, S.;  Abbafati, C.;  Abera, S. F.;  Abraham, J. P.;  Abu-Rmeileh, N. M.;  Achoki, T.;  AlBuhairan, F. S.;  Alemu, Z. A.;  Alfonso, R.;  Ali, M. K.;  Ali, R.;  Guzman, N. A.;  Ammar, W.;  Anwari, P.;  Banerjee, A.;  Barquera, S.;  Basu, S.;  Bennett, D. A.;  Bhutta, Z.;  Blore, J.;  Cabral, N.;  Nonato, I. C.;  Chang, J. C.;  Chowdhury, R.;  Courville, K. J.;  Criqui, M. H.;  Cundiff, D. K.;  Dabhadkar, K. C.;  Dandona, L.;  Davis, A.;  Dayama, A.;  Dharmaratne, S. D.;  Ding, E. L.;  Durrani, A. M.;  Esteghamati, A.;  Farzadfar, F.;  Fay, D. F.;  Feigin, V. L.;  Flaxman, A.;  Forouzanfar, M. H.;  Goto, A.;  Green, M. A.;  Gupta, R.;  Hafezi-Nejad, N.;  Hankey, G. J.;  Harewood, H. C.;  Havmoeller, R.;  Hay, S.;  Hernandez, L.;  Husseini, A.;  Idrisov, B. T.;  Ikeda, N.;  Islami, F.;  Jahangir, E.;  Jassal, S. K.;  Jee, S. H.;  Jeffreys, M.;  Jonas, J. B.;  Kabagambe, E. K.;  Khalifa, S. E.;  Kengne, A. P.;  Khader, Y. S.;  Khang, Y. H.;  Kim, D.;  Kimokoti, R. W.;  Kinge, J. M.;  Kokubo, Y.;  Kosen, S.;  Kwan, G.;  Lai, T.;  Leinsalu, M.;  Li, Y.;  Liang, X.;  Liu, S.;  Logroscino, G.;  Lotufo, P. A.;  Lu, Y.;  Ma, J.;  Mainoo, N. K.;  Mensah, G. A.;  Merriman, T. R.;  Mokdad, A. H.;  Moschandreas, J.;  Naghavi, M.;  Naheed, A.;  Nand, D.;  Narayan, K. M.;  Nelson, E. L.;  Neuhouser, M. L.;  Nisar, M. I.;  Ohkubo, T.;  Oti, S. O.;  Pedroza, A.;  Prabhakaran, D.;  Roy, N.;  Sampson, U.;  Seo, H.;  Sepanlou, S. G.;  Shibuya, K.;  Shiri, R.;  Shiue, I.;  Singh, G. M.;  Singh, J. A.;  Skirbekk, V.;  Stapelberg, N. J.;  Sturua, L.;  Sykes, B. L.;  Tobias, M.;  Tran, B. X.;  Trasande, L.;  Toyoshima, H.;  van de Vijver, S.;  Vasankari, T. J.;  Veerman, J. L.;  Velasquez-Melendez, G.;  Vlassov, V. V.;  Vollset, S. E.;  Vos, T.;  Wang, C.;  Wang, X.;  Weiderpass, E.;  Werdecker, A.;  Wright, J. L.;  Yang, Y. C.;  Yatsuya, H.;  Yoon, J.;  Yoon, S. J.;  Zhao, Y.;  Zhou, M.;  Zhu, S.;  Lopez, A. D.;  Murray, C. J.; Gakidou, E., Global, regional, and national prevalence of overweight and obesity in children and adults during 1980-2013: a systematic analysis for the Global Burden of Disease Study 2013. Lancet (London, England) 2014, 384 (9945), 766-81.
  15. Kang, H. T.; Shim, J. Y.;  Lee, H. R.;  Park, B. J.;  Linton, J. A.; Lee, Y. J., Trends in prevalence of overweight and obesity in Korean adults, 1998-2009: the Korean National Health and Nutrition Examination Survey. Journal of epidemiology 2014, 24 (2), 109-16.
  16. Rice, M. S.; Tamimi, R. M.;  Bertrand, K. A.;  Scott, C. G.;  Jensen, M. R.;  Norman, A. D.;  Visscher, D. W.;  Chen, Y.-Y.;  Brandt, K. R.;  Couch, F. J.;  Shepherd, J. A.;  Fan, B.;  Wu, F.-F.;  Ma, L.;  Collins, L. C.;  Cummings, S. R.;  Kerlikowske, K.; Vachon, C. M., Does mammographic density mediate risk factor associations with breast cancer? An analysis by tumor characteristics. Breast cancer research and treatment 2018, 170 (1), 129-141.
  17. Rice, M. S.; Bertrand, K. A.;  VanderWeele, T. J.;  Rosner, B. A.;  Liao, X.;  Adami, H. O.; Tamimi, R. M., Mammographic density and breast cancer risk: a mediation analysis. Breast cancer research : BCR 2016, 18 (1), 94.
  18. Choi, E.; Lee, Y. Y.;  Suh, M.;  Lee, E. Y.;  Mai, T. T. X.;  Ki, M.;  Oh, J. K.;  Cho, H.;  Park, B.;  Jun, J. K.;  Kim, Y.; Choi, K. S., Socioeconomic Inequalities in Cervical and Breast Cancer Screening among Women in Korea, 2005-2015. Yonsei Med J 2018, 59 (9), 1026-1033.

Reviewer 2 Report

Short summary

The authors have conducted a study in a large population of Asian women participating in breast cancer screening programs. Goal was to determine the association of known risk factors for developing breast cancer and the risk of actually developing breast cancer between different BIRADS density categories. This could be very useful to individualise the intensity of the screening programs.

Comments

Section ‘introduction:

-Line 46-50: ‘Most of the previous studies that investigated the interaction effect between breast density and the other risk factors on breast cancer development were based on the presence or absence of the known risk factors 8-13. Considering the preventive point, whether the effects of known breast cancer risk factors would be different (or modified) by breast density needs to be presented to identify interaction’. To me it is unclear what the authors are trying to say.

-Could the authors make clear what the ultimate goal of this research is? Is it prevention/lifestyle coaching on individual basis, is it designing a risk-based screening program?

Section: ‘methods’

-Could the authors comment on the ‘follow-up time of the study. Do they think that >7 years is enough? And what consequences could this have for the results? I would like to see more of this in the ‘discussion’.

-What is the reason that the risk factors ‘age at first birth’, ‘breast feeding’ and ‘parity’ were not investigated? These are known risk factors for breast cancer and also have shown to have a correlation with breast density.

Section `Results`:

-page 2 line 67: increasing age at menopause is described as giving an increased risk of developing breast cancer across all densitiy classes. However, line 72 and 74 seem to describe the opposite. In line 72, postmenopausal status is possibly meant?

-In table 1 I am puzzled about some of the numbers.

For example 80% never used oral contraceptives, more than 94% never smoked, en almost 77% did not use alcohol at the time patients filled out the self-reported questionnaires. These numbers are quite different from what I would expect, in the European population anyhow. Could the authors elaborate on that?

Section `Discussion`:

-line 106/107 states that age and age at menopause showed a heterogenous association with breast cancer across density categories. However, in the `results` section it is stated that advanced age at menopause increased breast cancer risk in all BI-RADS categories. To me that seems the opposite.

- Is miss one limitation in the discussion. A few risk factors can very in a persons life, for example BMI and use of hormonal medication. The study only knows about the status of these factors at 1 time point. This should be added.

-The authors mention in the conclusion that further research is needed. Do the authors have a suggestion (for the ‘discussion’ section) what this research should focus on/ how it should be performed. Is longer follow-up? Or is other information lacking? And should we focus on using the information for prevention or for designing a risk-based screening program?

Author Response

Thank you for your valuable comments.

We have revised the paper in accordance with your comments. The revisions are summarized below and highlighted in the manuscript. In addition, we have again opted for proofreading service intensively from an academic editing company. We have attached the respective certificate.

Short summary

The authors have conducted a study in a large population of Asian women participating in breast cancer screening programs. Goal was to determine the association of known risk factors for developing breast cancer and the risk of actually developing breast cancer between different BIRADS density categories. This could be very useful to individualise the intensity of the screening programs.

Response: Thank you for your comment. We revised the manuscript according to your comments below.

Comments

Section ‘introduction:

-Line 46-50: ‘Most of the previous studies that investigated the interaction effect between breast density and the other risk factors on breast cancer development were based on the presence or absence of the known risk factors 8-13. Considering the preventive point, whether the effects of known breast cancer risk factors would be different (or modified) by breast density needs to be presented to identify interaction’. To me it is unclear what the authors are trying to say.

Response: Thank you for your comment. From the preventive standpoint, the variable not amenable to intervention is usually regarded as the effect modifier and the exposure that can be eliminated or prevented or novel potential risk factors are regarded as the independent variables1. Previous studies considered other known risk factors as effect modifiers and breast density as an independent variable of interest. In this study, we considered breast density as an effect modifier and the other risk factors as independent variables of interest. To clarify we included following comment:

“Most previous studies that investigated the interaction effect between breast density and other risk factors on breast cancer development considered other risk factors as effect modifiers and breast density as an independent variable of interest2-7. Considering a preventive approach in treatment, whether the effects of other breast cancer risk factors differ by breast density (as an effect modifier)1 needs to be assessed.” (line 47-50)

-Could the authors make clear what the ultimate goal of this research is? Is it prevention/lifestyle coaching on individual basis, is it designing a risk-based screening program?

Response: We gave more a clear description of the goal of this study as follows:

“Therefore, in this study, we comprehensively examined the interaction effect between breast density measured during mammographic screening and various risk factors on breast cancer development by assessing the heterogeneity of the effect of these risk factors based on breast density in a large East Asian population for individualized risk assessment within the screening setting.” (line 57-61)

Section: ‘methods’

-Could the authors comment on the ‘follow-up time of the study. Do they think that >7 years is enough? And what consequences could this have for the results? I would like to see more of this in the ‘discussion’.

Response: To clarify, we added discussion of the methods as follows:

“The follow-up was considered from the date of breast cancer screening during the year of 2009–2010 up to December 31, 2018, date of death, or date of any cancer incidence (based on which date came first).” (line 295-297)

Considering the start of follow-up, at least 8 years of follow-up time is guaranteed. Thus, we changed >7 years to >8 years. In addition, for the discussion, we added the following description of the follow-up period:

“The follow-up time of ≥8 years would not be enough to identify all breast cancers. Breast cancer risk assessment estimates mostly 5-year risk but long-term risk assessment for 10 years or more is also needed8. Thus, the follow-up period of this study represents an intermediate time point.” (line 241-244)

-What is the reason that the risk factors ‘age at first birth’, ‘breast feeding’ and ‘parity’ were not investigated? These are known risk factors for breast cancer and also have shown to have a correlation with breast density.

Response: Due to the lack of information, risk factors such as age at first birth, breast feeding, and parity could not be considered. We included it as one of our limitations as follows:

“Regarding risk factors, due to a lack of information, some important risk factors for breast cancer, such as parity-related factors, could not be considered. Therefore, the residual confounding effect of unaccounted variables or variables with broad categories may affect the results.” (line 229-232)

Section `Results`:

-page 2 line 67: increasing age at menopause is described as giving an increased risk of developing breast cancer across all densitiy classes. However, line 72 and 74 seem to describe the opposite. In line 72, postmenopausal status is possibly meant?

Response: We are sorry for the confusion. Regarding your comment, we meant postmenopausal status itself. To clarify, we corrected the following sentence:

“Advanced age at menarche, post-menopausal status, and no use of hormone replacement therapy after menopause decreased breast cancer risk; a family history of breast cancer increased the risk irrespective of density category” (line 81-83)

-In table 1 I am puzzled about some of the numbers.

Response: Sorry for the all typos. We corrected all the typos that affected the numbers and percentages in Table 1. Please refer to the revised Table 1.

For example 80% never used oral contraceptives, more than 94% never smoked, en almost 77% did not use alcohol at the time patients filled out the self-reported questionnaires. These numbers are quite different from what I would expect, in the European population anyhow. Could the authors elaborate on that?

Response: Based on the Korea National Health & Nutrition Examination Survey, the current smoking rate was around 6% and drinking, defined as 1 or more drinks per month during the last year, was around 40% in women aged ≥19 years in the year 2009–2010 (https://knhanes.cdc.go.kr/knhanes/eng/index.do)9. In this study, smoking was defined as lifetime ever-smoking and drinking was defined based on the mean number of days of alcohol per week during the last year (once or more: drinking, less than once per week: no drinking). Thus, compared with the nationwide survey, the results would be valid. In addition, another study estimated the prevalence of never-smokers in Korean women as 95.8%10, supporting the prevalence of smoking and drinking in this study population.

Regarding oral contraceptive use, another study in a Korean population estimated the proportion of ever use as 18% (82% were never users)11. We mentioned the definition of smoking and drinking in the Methods and Tables. In addition, the prevalence of smoking, drinking, and oral contraceptive use was described in the Discussion as follows:  

“The questionnaire for health examination included smoking history in one’s lifetime, mean days of drinking per week during the last 1 year, and mean days of physical activity during the last one week.” (line 283-285)

“It could be attributed to the low prevalence of ever-smoking in Korean women in this study’s subjects, which was similar to that reported in previous studies9, 10” (line 197-198)

“Lower use of oral contraceptives in the Korean population may be the cause of the non-significant association11.” (line 206-207)

“In Korean women, the prevalence of drinking, defined as ≥1 drink of alcohol per month during the last year, was around 40%9 and the average alcohol consumption was 8 g/day12. (line 210-211)”

Section `Discussion`:

-line 106/107 states that age and age at menopause showed a heterogenous association with breast cancer across density categories. However, in the `results` section it is stated that advanced age at menopause increased breast cancer risk in all BI-RADS categories. To me that seems the opposite.

Response: We are sorry for the confusion. Although advanced age at menopause increased breast cancer risk in all BI-RADS categories, the P-heterogeneity=0.009, suggesting a statistically heterogenous association. To clarify we added the following:

“Advanced age at menopause increased breast cancer risk in all BI-RADS categories; this was significantly more prominent in participants with BI-RADS density category 1~” (line 75-77)

- Is miss one limitation in the discussion. A few risk factors can very in a persons life, for example BMI and use of hormonal medication. The study only knows about the status of these factors at 1 time point. This should be added.

Response: Thank you for your valuable comment. We added it as one of the limitations as follows:

“In addition, several risk factors, such as BMI, oral contraceptive use, or hormone replacement therapy use, can be changed during one’s lifetime; however, we considered these risk factors at a single time point at their screening during the year of 2009–2010.” (line 233-236)

-The authors mention in the conclusion that further research is needed. Do the authors have a suggestion (for the ‘discussion’ section) what this research should focus on/ how it should be performed. Is longer follow-up? Or is other information lacking? And should we focus on using the information for prevention or for designing a risk-based screening program?

Response: We clarified future research fields in the Conclusion as follows:

“Among various hypotheses regarding the mechanism of effect of breast density on breast cancer risk, this research focused on the interaction effect. However, studies have suggested the mediation effect of breast density on the association between known risk factors and breast cancer13, 14. The role of breast density in breast cancer risk in Asian women, including its mediation effect, needs to be considered in future studies.” (line 308-312)

References

  1. Szklo, M.; Nieto, F. J., Epidemiology: beyond the basics. Jones & Bartlett Publishers: 2014.
  2. Yaghjyan, L.; Colditz, G. A.;  Rosner, B.; Tamimi, R. M., Mammographic breast density and breast cancer risk: interactions of percent density, absolute dense, and non-dense areas with breast cancer risk factors. Breast cancer research and treatment 2015, 150 (1), 181-9.
  3. Maskarinec, G.; Nakamura, K. L.;  Woolcott, C. G.;  Conroy, S. M.;  Byrne, C.;  Nagata, C.;  Ursin, G.; Vachon, C. M., Mammographic density and breast cancer risk by family history in women of white and Asian ancestry. Cancer causes & control : CCC 2015, 26 (4), 621-6.
  4. Conroy, S. M.; Woolcott, C. G.;  Koga, K. R.;  Byrne, C.;  Nagata, C.;  Ursin, G.;  Vachon, C. M.;  Yaffe, M. J.;  Pagano, I.; Maskarinec, G., Mammographic density and risk of breast cancer by adiposity: an analysis of four case-control studies. Int J Cancer 2012, 130 (8), 1915-24.
  5. Maskarinec, G.; Dartois, L.;  Delaloge, S.;  Hopper, J.;  Clavel-Chapelon, F.; Baglietto, L., Tumor characteristics and family history in relation to mammographic density and breast cancer: The French E3N cohort. Cancer Epidemiol 2017, 49, 156-160.
  6. Woolcott, C. G.; Koga, K.;  Conroy, S. M.;  Byrne, C.;  Nagata, C.;  Ursin, G.;  Vachon, C. M.;  Yaffe, M. J.;  Pagano, I.; Maskarinec, G., Mammographic density, parity and age at first birth, and risk of breast cancer: an analysis of four case-control studies. Breast cancer research and treatment 2012, 132 (3), 1163-71.
  7. Yaghjyan, L.; Tamimi, R. M.;  Bertrand, K. A.;  Scott, C. G.;  Jensen, M. R.;  Pankratz, V. S.;  Brandt, K.;  Visscher, D.;  Norman, A.;  Couch, F.;  Shepherd, J.;  Fan, B.;  Chen, Y. Y.;  Ma, L.;  Beck, A. H.;  Cummings, S. R.;  Kerlikowske, K.; Vachon, C. M., Interaction of mammographic breast density with menopausal status and postmenopausal hormone use in relation to the risk of aggressive breast cancer subtypes. Breast cancer research and treatment 2017, 165 (2), 421-431.
  8. Brentnall, A. R.; Cuzick, J.;  Buist, D. S. M.; Bowles, E. J. A., Long-term Accuracy of Breast Cancer Risk Assessment Combining Classic Risk Factors and Breast Density. JAMA Oncol 2018, 4 (9), e180174-e180174.
  9. Web page of Korean National Health & Nutrition Examination Survey [Available at: https://knhanes.cdc.go.kr/knhanes/eng/index.do]. Last accessed 09 May 2020.
  10. Park, S.; Jee, S. H.;  Shin, H. R.;  Park, E. H.;  Shin, A.;  Jung, K. W.;  Hwang, S. S.;  Cha, E. S.;  Yun, Y. H.;  Park, S. K.;  Boniol, M.; Boffetta, P., Attributable fraction of tobacco smoking on cancer using population-based nationwide cancer incidence and mortality data in Korea. BMC cancer 2014, 14, 406.
  11. Park, B.; Park, S.;  Shin, H. R.;  Shin, A.;  Yeo, Y.;  Choi, J. Y.;  Jung, K. W.;  Kim, B. G.;  Kim, Y. M.;  Noh, D. Y.;  Ahn, S. H.;  Kim, J. W.;  Kang, S.;  Kim, J. H.;  Kim, T. J.;  Kang, D.;  Yoo, K. Y.; Park, S. K., Population attributable risks of modifiable reproductive factors for breast and ovarian cancers in Korea. BMC cancer 2016, 16, 5.
  12. Park, S.; Shin, H. R.;  Lee, B.;  Shin, A.;  Jung, K. W.;  Lee, D. H.;  Jee, S. H.;  Cho, S. I.;  Park, S. K.;  Boniol, M.;  Boffetta, P.; Weiderpass, E., Attributable fraction of alcohol consumption on cancer using population-based nationwide cancer incidence and mortality data in the Republic of Korea. BMC cancer 2014, 14, 420.
  13. Rice, M. S.; Tamimi, R. M.;  Bertrand, K. A.;  Scott, C. G.;  Jensen, M. R.;  Norman, A. D.;  Visscher, D. W.;  Chen, Y.-Y.;  Brandt, K. R.;  Couch, F. J.;  Shepherd, J. A.;  Fan, B.;  Wu, F.-F.;  Ma, L.;  Collins, L. C.;  Cummings, S. R.;  Kerlikowske, K.; Vachon, C. M., Does mammographic density mediate risk factor associations with breast cancer? An analysis by tumor characteristics. Breast cancer research and treatment 2018, 170 (1), 129-141.
  14. Rice, M. S.; Bertrand, K. A.;  VanderWeele, T. J.;  Rosner, B. A.;  Liao, X.;  Adami, H. O.; Tamimi, R. M., Mammographic density and breast cancer risk: a mediation analysis. Breast cancer research : BCR 2016, 18 (1), 94.
